# Zero-Bias Visible to Near-Infrared Horizontal p-n-p TiO_2_ Nanotubes Doped Monolayer Graphene Photodetector

**DOI:** 10.3390/molecules24101870

**Published:** 2019-05-15

**Authors:** Zehua Huang, Chunhui Ji, Luhua Cheng, Jiayue Han, Ming Yang, Xiongbang Wei, Yadong Jiang, Jun Wang

**Affiliations:** School of Optoelectronic Science and Engineering State Key Laboratory of Electronic Thin Films and Integrated Devices, University of Electronic Science and Technology of China, Chengdu 610054, China; jadeofuestc@126.com (Z.H.); jichunhui_uestc@163.com (C.J.); cheng.luhua@foxmail.com (L.C.); hanjiayue_uestc@163.com (J.H.); yangming932@163.com (M.Y.); weixiongbang@uestc.edu.cn (X.W.); jiangyd@uestc.edu.cn (Y.J.)

**Keywords:** graphene, TiO_2_ nanotubes, broadband, zero-bias, p-n-p junction

## Abstract

We present a p-n-p monolayer graphene photodetector doped with titanium dioxide nanotubes for detecting light from visible to near-infrared (405 to 1310 nm) region. The built-in electric field separates the photo-induced electrons and holes to generate photocurrent without bias voltage, which allows the device to have meager power consumption. Moreover, the detector is very sensitive to the illumination area, and we analyze the reason using the energy band theory. The response time of the detector is about 30 ms. The horizontal p-n-p device is a suitable candidate in zero-bias optoelectronic applications.

## 1. Introduction

As the most representative two-dimensional material, graphene has attracted a large amount of attention from researchers all over the world. Due to its high mobility, unique optical properties, being gapless at the Dirac point, and its excellent conductivity, graphene has become a potentially useful material for optoelectronic devices, such as light modulators [1], photodetectors [2,3,4], and photodiodes [5]. Limited by the bandgap, conventional silicon can only be operated below a 1.1 µm wavelength [6,7]. III–V semiconductors, such as GaAs(gallium arsenide), usually has high mobility and direct gap, but the high production cost, mainly because the materials used are rare, limits their application [8]. Graphene has no bandgap at the Dirac point, and this makes it an excellent candidate for broadband photodetectors, from UV to THz frequencies [9,10,11,12,13]. As for photodetectors, the photo-induced electronics and holes separate to the electrodes to form a photocurrent [14]. Combining graphene with nanoparticles, nanowires, and other thin films form heterojunctions [15,16,17,18,19,20], or forming a p-n junction by spilt gates [21] can be applied for separating the carriers. Graphene generally exhibits p-type in the atmosphere due to the influence of water vapor and oxygen. TiO_2_ nanotube is an n-type semiconductor with high chemical stability, a low price, and ease in preparation [22,23]. Zhang et al. used TiO_2_ and graphene heterojunction for UV detection [16,24]. We here reported a broadband horizontal p-n-p photodetector, which was fabricated by transferring single-layer graphene onto the surface of TiO_2_ nanotubes as shown in Figure 1a. The TiO_2_ nanotubes act as donors, and they dope the part of graphene on the nanotubes with electrons and form a horizontal p-n-p junction. Integrating the graphene with TiO2 nanotubes improved the photoresponse intensity relative to the monolayer graphene. Moreover, we use the energy band theory to analyze detectors that are very sensitive to the illuminated area. It was found that the graphene/TiO_2_ nanotubes photodetector works without external bias and it is expected to be a promising zero-bias broadband photodetector. In addition, the p-n-p junction makes the photodetector sensitive to the irradiation area, which means this kind of device is a candidate for dynamic object imaging device.

## 2. Experimental Section

### 2.1. Preparation of the Graphene/TiO_2_ Nanotubes Photodetector

The TiO_2_ nanotubes arrays were prepared on 0.1 mm–thick Ti sheets of 99.8% purity, (HeFei Ke Jing Materials Technology Co., LTD, HeFei, China). We reported in a previous article that TiO_2_ nanotubes were prepared by anodic oxidation method [25]. The thin Ti sheet was treated by anodic oxidation for 180 min with 0.5 wt % HF electrolyte aqueous solution, 60 V DC voltage, and 0.25 wt % NH_4_F/glycol organic electrolyte aqueous solution at room temperature. A total of 180 min thermal annealing in the air atmosphere at 450 °C was followed by anodization to form high ordered anatase TiO_2_ nanotubes arrays. After annealing, the titanium oxide nanotubes need to be carefully treated because the titanium oxide nanotubes are easily detached from the titanium sheet and are drawn into the body through the air [26,27,28]. Monolayer graphene synthesized by using a chemical vapor deposition (CVD) method [29] and was transferred to TiO_2_ nanotubes after depositing gold electrodes.

We used monolayer graphene synthesized on a copper foil with the CVD method to fabricate the photodetector. Transferring the graphene to the preset TiO_2_ nanotubes is necessary. After the graphene has been transferred to the substrate, it is brought into contact with the substrate by Van der Waals force. This force is much weaker than the force of the molecular bond. Despite the single-atom-thick nature, graphene is very compact—water and gas molecules do not readily penetrate monolayer graphene. Tan et al. used the monolayer graphene to keep their perovskite film away from the water [30]. Therefore, once the monolayer graphene was transferred to a flat substrate, the large-area monolayer graphene can strictly adhere to the substrate surface under the influence of atmospheric pressure. Unlike the smooth Si/SiO_2_ substrate, the surface of the TiO_2_ nanotubes manufactured by the anodic oxidation method is rugged, and the nanotubes have a hollow structure, as shown in Figure 1d. Due to the inconsistent tube length of the nanotubes, the surface of the nanotubes has undulations of several tens of nanometers, as shown in Figure 1e. The fluctuations of the nanotubes make it difficult for graphene to contact the substrate sufficiently. Since the nanotubes are hollow, and some of the nanotubes are convex, the graphene in many irregular regions is suspended. This is not conducive to the adhesion of graphene to the substrate. In addition to the hollow portion of the nanotube, many irregular areas also cause the graphene to float, which is not conducive to the adhesion of graphene to the substrate. At the same time, since some parts of the graphene are suspended, it is easy to cause damage to the graphene when the PMMA is removed. It is difficult to maintain the monolayer graphene smoothly and tightly on the TiO_2_ nanotubes as the graphene can fall off the TiO_2_ nanotubes after the dying and annealing treatment. To solve this problem, we first deposited gold on the TiO_2_ nanotubes to create a relatively flat surface, and then we transferred the monolayer graphene onto the substrate. The 60-nm gold layer, deposited with using electron beam evaporation can fill the holes of the hollow TiO_2_ nanotubes, as shown in Figure 1f and provided more contact surface. Isopropanol was used to remove the residual water between the graphene and TiO_2_ nanotubes. With the assistance of a flat gold surface, graphene can fit smoothly over the surface of the substrate.

### 2.2. Characterization Methods

A scanning electron microscope (SEM; FEI-Inspect F50, Thermo Fisher Scientific, Watham, MA, USA) was used to investigate the morphology of the graphene and the TiO_2_ nanotubes. Raman spectroscopy (RENISHAW inVia Raman Microscope, Renishaw, UK) was used to identify the number of layers of graphene. Atomic force microscopy (AFM, Asylum Research MFP-3D, Oxford Instruments, UK) characterized the surface morphologies of the TiO_2_ nanotubes. The current–time curves were measured with the Keithley 2636B sourcemeter(Tektronix Inc, Beaverton, OR, USA).

## 3. Results and Discussion

TiO_2_ grown on the titanium sheet provides many unique characteristics, such as a highly ordered one-dimensional structure, high chemical stability, low cost, and tunable morphology [31,32,33]. The tubular structure of TiO_2_ nanotubes can enhance the interaction with light. Figure 1a shows the photodetector structure schematic. Monolayer graphene was transferred onto the TiO_2_ nanotubes with gold electrodes. Figure 1b is the Raman spectrum of the graphene on the TiO_2_ nanotubes and Si/SiO_2_ substrate with 514 nm excitation laser. The 2D peak at 2697.53 cm^−1^ is significantly higher than the G peak at 1579.8 cm^−1^,which means the graphene on the nanotubes is a single layer. Due to the unevenness of the TiO_2_ nanotubes, the graphene is still wrinkled on a small scale after being transferred to the substrate, as shown in Figure 1c. The small-scale wrinkle leads to the 2D and G peak intensity ratio of I_2D_/I_G_ < 2. the 2D peak of graphene on the Si/SiO_2_ substrate appears at 2685.3 cm^−1^. The varieties fit well with previous research results [34], as the number of graphene layers increases, the 2D peak gets a blue shift. The pleats of graphene cause this blue shift. The diameters of the hollow TiO_2_ nanotubes arrange from 90 to 160 nm.

Figure 2a depicts the time–current curves of the photodetector with different bias voltages for the TiO_2_ nanotubes/graphene device. The device can work under low bias voltages and even no bias. The laser frequency is 405 nm, and the power density of the laser is 3.63 mW/cm^2^. The current responsivity R(I) = I_photo_/P, where R(I) is the current responsivity, I_photo_ is the photocurrent, P is the effective illumination power [35,36]. Moreover, P = A∙P_0_, where A is the active illumination area and P_0_ is the optical power density. The optical power density of the 405 nm laser is P_0_ = 3.63 mW/cm^2^, and the active illumination area is about A = 0.34 mm^2^. The detector generates a 55 nA photocurrent with different bias voltages, and the computational current responsivity is about 4.5 mA/W. To compare with a pure monolayer graphene and TiO_2_ nanotubes, we tested the photoresponse of pure graphene and TiO_2_ nanotubes without graphene as shown in Figure 2d,e. The TiO_2_ nanotubes without graphene has no photoresponse at 405 nm laser illuminating as shown in Figure 2e. The photocurrent of the graphene is less than 1 nA which is much weaker than the graphene/TiO_2_ nanotubes device. Although the carrier mobility of graphene is very high (∼1810 ± 710 cm^2^ /(V∙s) with isopropanol treatment at 300 K [37]), the lifetime of the photo-induced carriers is quite short. As it is a one-atom layer, the monolayer graphene has low light absorption. The relaxation time of graphene is short and this does not allow carrier multiplication [38]. With the introduction of the built-in electric field, the carriers separated in the small built-in electric field area increases the carriers collection efficiency to acquire higher photocurrent and lower response time as shown in Figure 2b–d. The response time of the photodetector was tested using an oscilloscope with a 405-nm laser illumination. The rise time T_r_ ≈ 30 ms and fall time T_f_ ≈ 37 ms are shown in Figure 2c.

With the same laser power, the photocurrent of the 405 nm laser is more significant than other frequencies, which means that the TiO_2_ nanotubes provide photo-induced carriers for the monolayer graphene. The characteristic of the zero bandgaps of graphene makes the detection frequency of the photodetector range from visible to near-infrared, as shown in Figure 2b. That means the graphene/TiO_2_ nanotubes photodetector is a promising candidate for the broadband photodetector. Figure 3a shows the structure of the photodetector with the light illuminating on only one p-n junction region of the graphene.

When the light was illuminating on the different side of the graphene, the photocurrent would be positive or negative. Since graphene usually adsorbs some moisture, oxygen, etc., on the surface, it usually shows p-type as shown in Figure 4a. After the p-type graphene was transferred on the n-type TiO_2_ nanotubes, the nanotubes acted as a donor and doped the middle region of the graphene with electrons. The monolayer graphene was divided into three states, and a p-n-p junction formed, as shown in Figure 3a. The device can be divided into three areas, marked as area 1–3. Area 1 and 2 are the regions where the graphene and TiO_2_ nanotubes are separated by gold. As the light illuminates the junction, light-induced electronhole pairs are separated by the built-in electric field and then captured by the external circuit to form a photocurrent. Because the graphene on the TiO_2_ nanotubes is wrinkled randomly, the character of the graphene on the opposite side will not be the same. The direction of the built-in electric field at both sides are opposite. If the light illuminates only one side of the graphene (p-n or n-p junction), a positive response I_1_ is generated when the built-in electric field is consistent with the direction of the external electric field, as shown in Figure 3b. When the built-in electric field is opposite to the direction of the external electric field, a negative response I_2_ occurs, as shown in Figure 3c. The incident light is a 650 nm laser. When adjusting the incident spot to one side of the device, a photocurrent of about 7.8 nA was generated. When the laser spot was focused on the other side of the device, a photocurrent of about −7.2 nA was generated. In addition, when the laser spot focused on the middle or covered the entire device, almost no photocurrent was generated, as shown in Figure 3d. The mean value of the positive photocurrent and negative photocurrent is well matched in the current with light illuminated on the entire device, as shown in Figure 3e,f.

The band structure of the horizontal p-n-p junction is shown in Figure 4b. The built-in electronic field of the p-n and the n-p junction is opposite in direction. The effective photo-generated current is |I_1_ + I_2_| if the light illuminated the whole device. However, the photocurrent I_1_ and I_2_ are in the opposite direction. The photocurrent weakens each other in the two different areas. Therefore, if the light spot illuminating on the whole device, the photoresponse is much weaker than the light illuminating on only one side of the device, as shown in Figure 3b,c,e. The photocurrent is generated in the p-n junction region of graphene, which exceeds the junction region and produces a very weak photocurrent. This is why the device is susceptible to the illumination region. The two different photoresponse states, with the light illuminating on the different side of the device, make the photo device an excellent candidate for scanning image elements.

## 4. Conclusions

In conclusion, we demonstrated TiO_2_ nanotubes doped the horizontal p-n-p monolayer graphene photodetector. The photodetector is sensitive to visible light to near-infrared. The reason why the device is sensitive to the illumination area is analyzed from the energy band structure of the device. We get a broad spectrum zero-bias photodetector from 405 nm to 1310 nm. The built-in electric field makes the device work without bias, which is friendly to the environment and to sustainable development. This kind of horizontal p-n-p device will be a candidate in low power consumption optoelectronic applications.

## Figures and Tables

**Figure 1 molecules-24-01870-f001:**
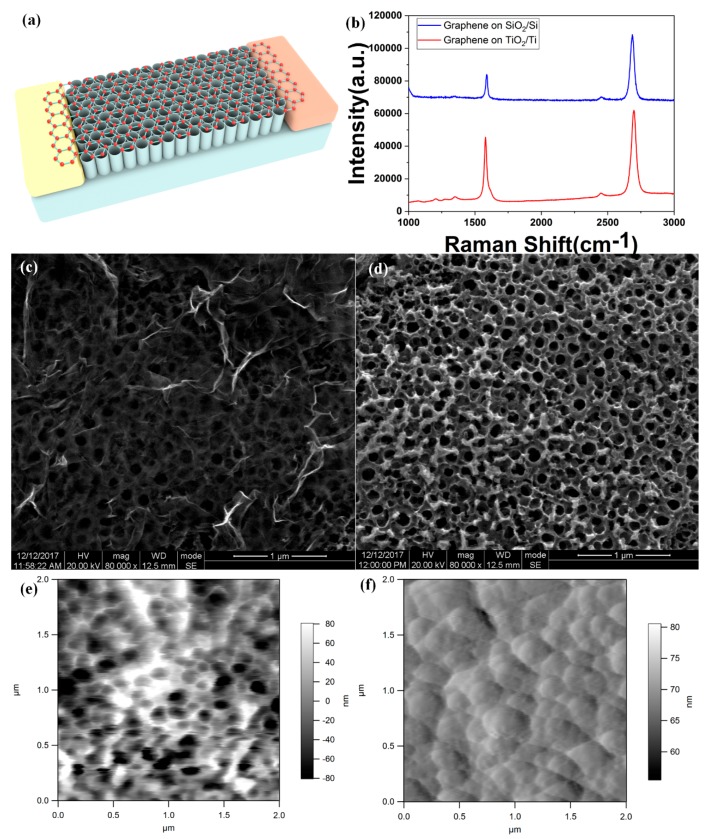
(**a**) Structural schematic of the graphene-TiO_2_ nanotubes heterojunction photodetector. (**b**) Raman shift spectrum of the graphene transferred on the TiO_2_ nanotube and Si/SiO_2_ substrate. (**c**) SEM image of the graphene transferred on the TiO_2_ nanotube. (**d**) Top view of the TiO_2_ nanotube before coating gold by SEM. (**e**) AFM(atomic force microscopy) of the TiO_2_ nanotubes. (**f**) AFM of the gold layer on the TiO_2_ nanotubes.

**Figure 2 molecules-24-01870-f002:**
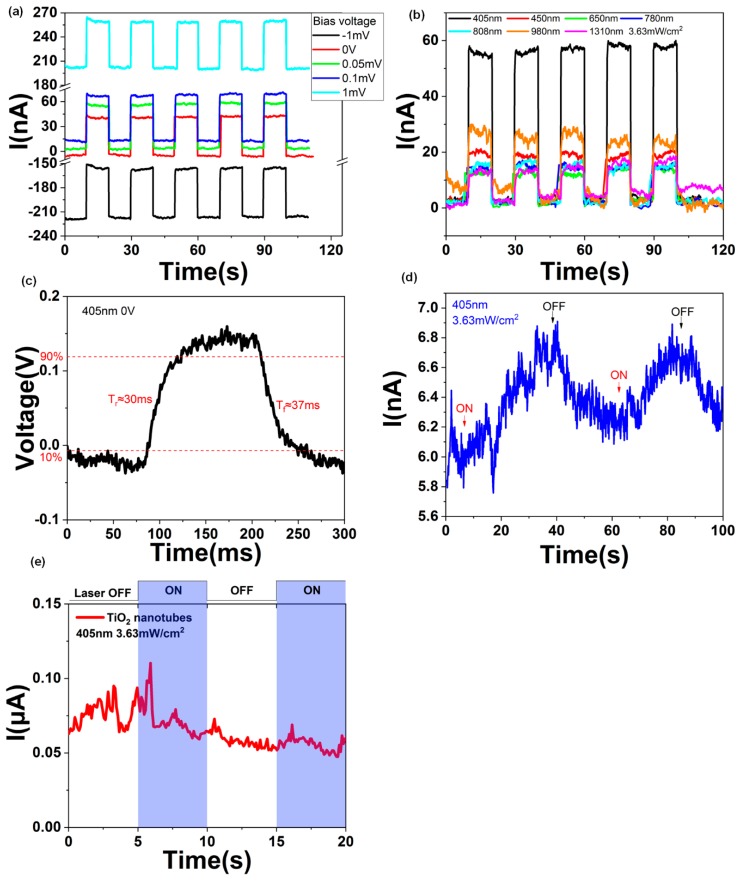
(**a**) Current and time curves with different bias voltage under 405 nm laser irradiating of the TiO_2_ nanotubes/graphene device. (**b**) Current and time curves under different laser irradiating of the TiO_2_ nanotubes/graphene device. (**c**) The response time of the graphene/TiO_2_ nanotubes photodetector under 405 nm laser irradiating without bias of the TiO_2_ nanotubes/graphene device. (**d**) Current and time curves of monolayer graphene on SiO_2_/Si under 405 nm laser irradiating. (**e**) Current and time curves of TiO_2_ nanotubes without graphene on it under 405 nm laser irradiating. The bias voltage is 0.2 V.

**Figure 3 molecules-24-01870-f003:**
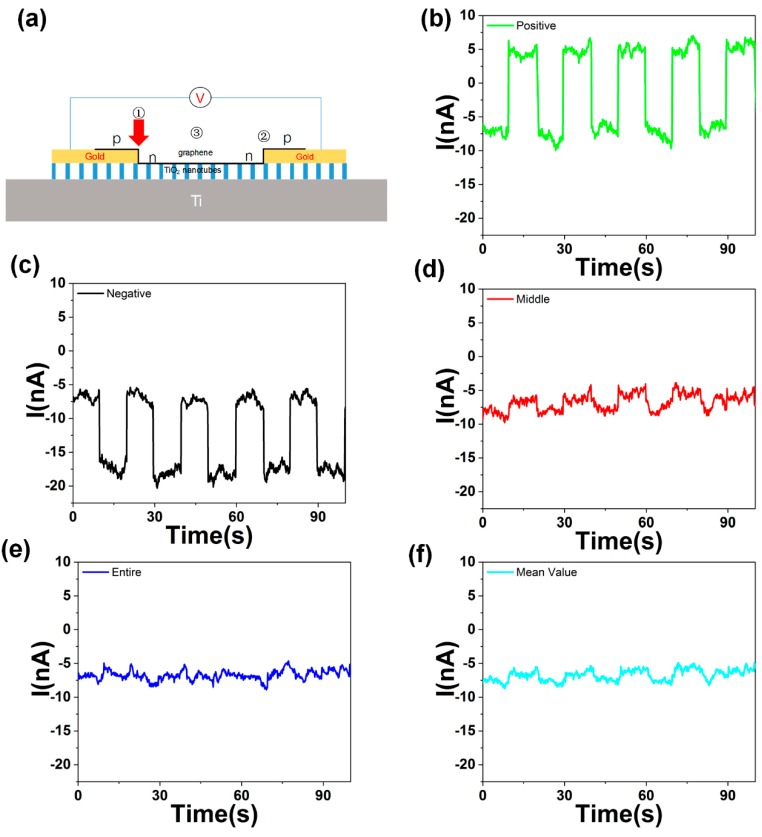
(**a**) Schematic of the photodetector. The monolayer graphene divided into three parts by the TiO_2_ nanotubes and formed a horizontal p-n-p junction. The current changes when the laser focuses on the different area of the photodetector. (**b**) Positive photoresponse of the device when light illuminating on area 1. (**c**) Negative photoresponse of the device when light illuminating on area 2. (**d**) Photoresponse with the laser illuminated in the middle area 3 of the device. (**e**) Photoresponse with the laser illuminated on the entire device. (**f**) Mean value of the positive and negative photocurrent with 650 nm light illuminating.

**Figure 4 molecules-24-01870-f004:**
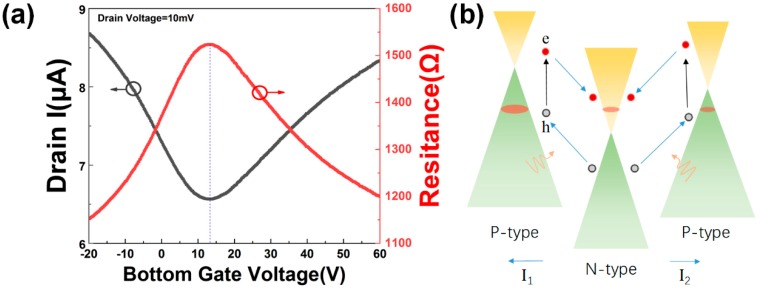
(**a**) Transfer characteristic curve of bottom gate graphene FET(field effect transistor) on Si/SiO_2_ substrate. The monolayer graphene transferred on the Si/SiO_2_ exhibits p-type. (**b**) Band structure of the horizontal p-n-p junction, blue arrows describe the movement of photo-induced electrons (red balls) and holes (gray balls).

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
