# Peer review of "Zero-Bias Visible to Near-Infrared Horizontal p-n-p TiO2 Nanotubes Doped Monolayer Graphene Photodetector"

_molecules, 2019, doi:10.3390/molecules24101870_

Round 1

Reviewer 1 Report

This manuscript reports on a TiO2 nanotube photodetector with a monolayer graphene which covers a wavelength range from 405 nm to 808 nm.  The authors showed material properties (SEM, Raman spectroscopy, AFM) and photodetection characteristics (time-current curves). They claimed the fabricated detector can work in photovoltaic mode by taking advantage of built-in electric field generated at graphene-TiO2 p-n junction. In my opinion, this manuscript is interesting to the readers of Molecules and has merits. Indeed, I have several concerns and would require revisions to clarify some points in the manuscript.

1. It is not clear throughout the manuscript that what is the motivation of combining graphene layer with TiO2 nanotube array. Graphene is not the only material which can form p-n junction with TiO2. Why it has to be graphene? In other words, it is not clear how graphene layer can improve photodetection performance of a TiO2 array (e.g. any enhancement of photodetection at infrared?).

2. The authors claim the detector can work at near-infrared (as emphasized in the title). However, there is not much discussion regarding the device performance in this wavelength regime. Most of results are based on photocurrent measurements at 405 nm.

3. What would be the takeaways/conclusions of time-current measurements? Such measurements can be use to show high-speed performance, which is not the focus of this manuscript. I would suggest that the authors can consider to include spectral response, which can give a clear signature of broad-band photodetection. Please justify.

4. The authors used graphene-on-SiO2/Si as a control. In my opinion, it would be more appropriate to use bare TiO2 nanotube array. The ideal is to use graphene to improve the detector performance of bare TiO2 nanotube array but not the other way around. Please explain and justify.

5. It claims that the responsivity at 405 nm is 4.5mA/W. How was this number calculated? Please indicate the active device area (or graphene area) and the laser spot size. Additionally, TiO2 can also absorb at 405 nm. How to distinguish the photocurrent contribution from either graphene or TiO2 nanotubes? I would suggest that the authors can refer to some previous studies on nanostructured array based photodetectors in which their device characterizations and design strategies are clearly discussed: (1) Nano Letters 18 (12), 7901, 2018; (2) Nanotechnology 30 (4), 044002, 2018; (3) Nano Letters 19 (1), 582, 2019; (4) Nano Letters, 10.1021/acs.nanolett.8b04420.

6. In Figure 3, the authors show the photocurrent is position-dependent – higher current while the excitation is close to the junction and lower current while the excitation is around the center of the graphene layer. This means that the internal quantum efficiency would be low if illumination is near the center. Please note that in an actual detector array illumination area (or incident light source) normally covers the entire device. This position-dependent photoresponse characteristic might not be desired. Please justify.

In general, the study shown in the manuscript is systematic and interesting. If possible, I would like see their revisions.

Author Response

First, we are very grateful for reviewing this manuscript and your focus on our work. Those comments are very constructive and helpful for revising and improving this work.

Then we have made corrections according to your comments in the revised manuscript. Please see the detail responses in the attachment.

Reviewer 2 Report

See the report.

Author Response

(The authors gave the same response as above.)

Author Response

(The authors gave the same response as above.)

Round 2

Author Response

(The authors gave the same response as above.)
